# Thermal insulation of poly(methyl methacrylate) bone cement and hydroxyapatite coatings under induction heating of metal implants

**Robert Kamphof**[1,2]*, **Rob G. H. H. Nelissen**[1], **Giuseppe Cama**[2], **Bart G. C. W. Pijls**[1]

1 Department of Orthopaedics, Leiden University Medical Center, Leiden, The Netherlands, 2 CAM Bioceramics B.V., Leiden, The Netherlands

* r.kamphof@lumc.nl

## Abstract

Prosthetic joint infection after joint arthroplasty has serious consequences for patients. Finding new treatments against prosthesis infection is increasingly urgent due to the growing number of prosthetic joints and increasing prevalence of anti-microbial resistance. This study investigates the feasibility of non-contact induction heating to supplement debridement, antibiotics and implant retention using an implant model. Upper temperature limits for induction heating treatment are established, based on the presence or absence of biomaterials commonly associated with joint replacement implants. Titanium grade 5 coupons were heated using an induction device to 50, 70 and 90°C. Heat transfer through poly(methyl methacrylate) bone cement and hydroxyapatite coatings was studied, with poly(acrylic acid) gel phantoms serving as tissue mimic. Thermal doses delivered at the biomaterial-gel interface were quantified. Thermographic images supported the findings. Thermal doses (CEM43) were calculated to estimate damage to human bone tissue. Safe induction temperatures vary by implant configuration. Cemented implants can be heated up to 70–80°C without risk of mechanical failure or patient harm, depending on the thickness of the cement mantle. For uncemented and hydroxyapatite-coated implants, the temperature limit is 50°C. Since temperatures were measured at the metal-biomaterial interface, higher temperatures could be safe for implant sections farther away from the bone. Thus, non-contact induction heating is a safe treatment strategy for prosthetic joint infection in both modalities.

## 1. Introduction

Prosthetic joint infection (PJI) is a devastating adverse outcome of total joint replacement surgery [1–4]. PJI accounts for 1–2% of all outcomes in primary arthroplasty patients, but can be as high as 15% in mega-implants after tumour resection [5,6]. As one of the main reasons for revision surgery, PJI results in extended hospital stay,

which permits unrestricted use, distribution, and reproduction in any medium, provided the original author and source are credited.

**Data availability statement:** Images of thermal curves and thermographs for all study groups and target temperatures are available in S1 and S2 Figs. The starting temperatures, peak temperatures and CEM43 results for individual replicates can be found in S3 Table. 3D model files (.STL) of the templates used to create Ti6Al4V-PMMA constructs and to apply PAA gel layers are available on the Harvard Dataverse at https://doi.org/10.7910/DVN/KIKEAX and https://doi.org/10.7910/DVN/OZ0HDO.

**Funding:** This publication is part of the project DARTBAC (with project number NWA.1292.19.354) of the research programme NWA-ORC which is (partly) financed by the Dutch Research Council (NWO). B.G.C.W. Pijls received funding from ZonMW under Veni Grant 09150161810084. The funders had no role in study design, data collection and analysis, decision to publish, or preparation of the manuscript.

**Competing interests:** R.G.H.H. Nelissen and B.G.C.W. Pijls are listed as inventors on patents from Leiden University Medical Center regarding induction heating of metal implants. This article was written in collaboration with CAM Bioceramics B.V., situated in Leiden, The Netherlands. CAM Bioceramics B.V. is a contract development and manufacturing organisation that designs, develops and manufactures calcium phosphates for a range of medical applications. This does not alter our adherence to PLOS ONE policies on sharing data and materials.

increased patient morbidity and an increased burden on the healthcare sector [3,4]. Treatment of PJI is complicated by the formation of biofilms on the implant, which shield microbes from antibiotics and the host immune system [2,7]. Furthermore, rising antimicrobial resistance (AMR) decreases the chance of successful treatment using antibiotics [8].

The first treatment step in early PJI is Debridement, Antibiotics and Implant Retention (DAIR), a surgical procedure that combines antibiotic treatment with mechanical cleaning of the implant. DAIR is effective at treating early post-operative and/or acute PJI, although there is a large variation in the reported success rate (c.a. 15–80% based on selection criteria and site of infection) [2,9–13]. If DAIR fails, more invasive treatment options are needed, such as one-stage or two-stage revision of the implant. Since these procedures are costly and a heavy burden on patients, optimising the success rate of DAIR is highly desirable. For this, non-contact induction heating (NCIH) may be used in the future.

NCIH utilises electromagnetic fields to deliver localised thermal damage to bacterial biofilms located on the surface of metal implants. This technique uses an induction coil to induce eddy currents in metal implants, producing heat to damage biofilm on the implant surface [14–18]. NCIH is promising since it can affect hard-to-reach places of the implant (e.g., screw holes), and it is possible to heat specific segments of an implant while leaving others unheated, reducing the risk of harming surrounding tissues [15]. Furthermore, we speculate that NCIH is potentially less susceptible to resistance development than drug-based treatments. NCIH has also been shown to increase the susceptibility of biofilms to antibiotics even if the induction treatment did not eradicate the infection on its own [14].

While higher temperatures are more effective at eradicating bacterial biofilms, increasing temperature comes with an increasing risk of thermonecrosis to the periprosthetic bone [15]. It is currently unknown to what extent the implant can be heated to maximise the destruction of biofilm while minimising the risk of tissue damage. This study aims to model heat transfer from the metal part of the implant to surrounding tissue in order to establish benchmarks for the temperatures that can be safely reached during NCIH, based on the implant configuration (cemented, uncemented or coated with hydroxyapatite).

Our model focuses on induction heating of the extra-osseus part of an implant such as a total knee or total hip arthroplasty (Fig 1). This approach offers two key advantages. First, the extra-osseus part of the implant is an attractive target for NCIH, since some antibiotics have low penetration into synovial fluid and the host immune response may be complicated by limited diffusion of oxygen and the rapid aggregation of bacteria in synovial fluid [19–21]. Second, by heating only the extra-osseus part, the chance of heating-related complications like implant loosening or tissue necrosis is minimised. Since this part of the implant is not in contact with tissue during DAIR, the primary risk comes from heat conducted down the implant to the implant-tissue interface.

The primary outcome of this study is the thermal dose delivered at the implant-tissue interface, expressed in CEM43 (equivalent minutes at 43°C). CEM43 is a unit

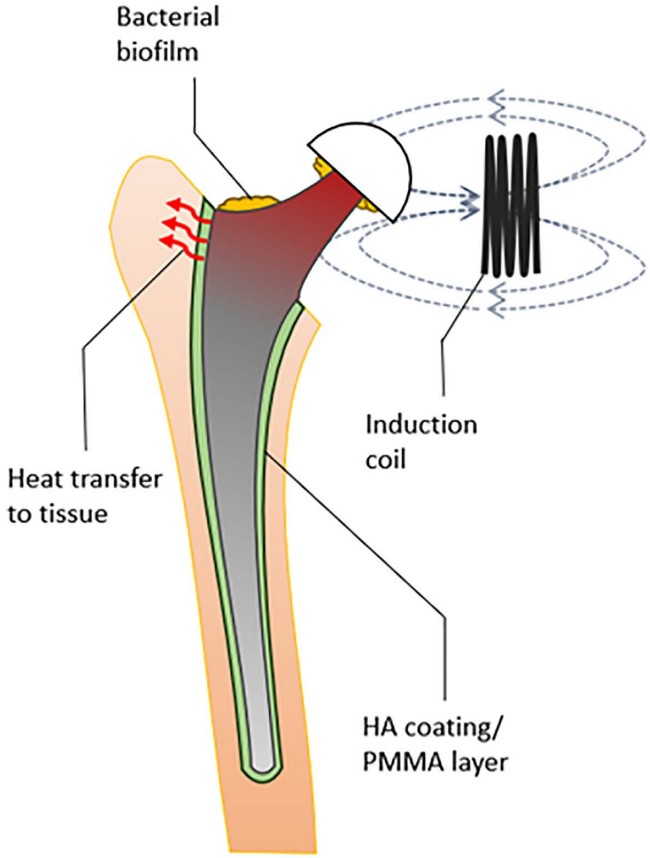

**Fig 1. Schematic showing an infected hip implant under NCIH treatment, where heat transfer to the bone is shielded by a layer of PMMA or HA.**

of thermal dose that has been shown to be an effective indicator of thermal damage [22,23]. Previously, a CEM43 greater than 16 minutes was found to be harmful to bone tissue [22]. We qualitatively and quantitatively assess the surface temperature of uncemented Ti6Al4V coupons, as well as coupons covered with HA coatings or a layer of bone cement. Additionally, the effect of heat absorption by tissue was modelled with the use of gel phantoms. Thermal doses experienced by the biomaterials' surfaces are used to estimate temperatures that can be reached safely.

## 2. Methods

To model the temperature generated by NCIH, constructs were created that incorporate the most common biomaterials found in total joint implants in a layer-by-layer approach. The metal implant, represented by a grade 5 titanium (Ti6Al4V) coupon (38 x 25 x 1mm), was heated to target temperatures of 50, 70 or 90°C. In previous research, these coupons were described in more detail, and used for mechanical testing after induction heating, and used to test the efficacy of NCIH on the eradication of biofilms [14,15,24,25].

To model the presence of biomaterials commonly associated with joint replacement implants, the coupons were partially covered by a layer of poly(methyl methacrylate) (PMMA) bone cement of 1, 3 or 5mm thickness or a coating of HA (50μm thick). These thicknesses are typical for clinical use [19–21,26]. To model the heat transfer to surrounding tissues, a gel phantom composed of poly(acrylic acid) (PAA) was produced. In total, 10 different study groups were investigated

([Fig 2]). The coupons were heated using an induction device and studied qualitatively, using infrared thermography, as well as quantitatively, using thermocouples.

## 2.1. Sample preparation & characterisation

### 2.1.1 Ti6Al4V-PMMA constructs.
Prior to adding layers of PMMA bone cement, a K-type thermocouple (Voltcraft TP-202) was glued onto the coupons at 25% distance from one of the short edges using cyanoacrylate superglue. This method was previously used to measure the temperature at the metal-cement interface [24]. The coupon was divided into two halves and the half of the coupon with thermocouple was subsequently covered in a layer of PMMA of thickness 1, 3 or 5 mm.

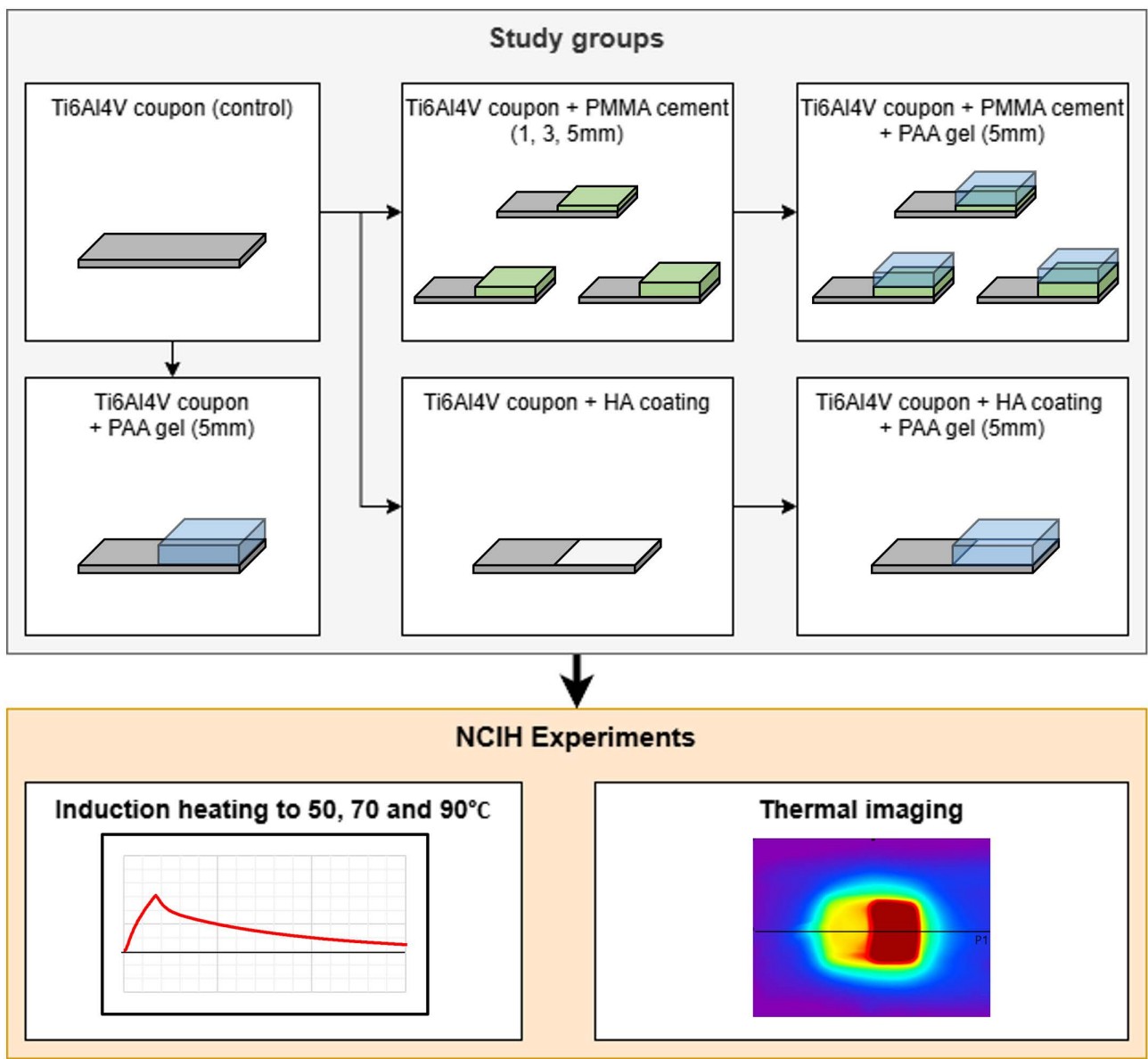

**Fig 2. A diagram outlining the general study design.**

Uniform PMMA layers were achieved using 3D-printed templates, which were made using a Neptune 3 Pro 3D printer from Elegoo (Fig 3). The printer was fitted with a 0.4 mm nozzle and 0.16 mm layer thickness was used. Poly-lactic acid (PLA) filament for printing was purchased from R3D via amazon.com. The 3D model files for printing were made using the free and open-source software Blender, version 3.3.1. These files (.STL and .BLEND) for templates with a cement thickness of 1, 3 and 5 mm are available at https://doi.org/10.7910/DVN/KIKEAX.

Palacos R + G PPMA bone cement (Heraeus Medical GmbH, Germany) was hand-mixed and inserted into the PLA templates as previously described [24]. After removing excess cement, Ti6Al4V coupons were pressed onto the hardening cement and secured using tape while the cement hardened. After hardening, the templates were removed to release the Ti6Al4V-PMMA constructs (Fig 4).

**2.1.2. T6Al4V-HA constructs.** HA powders and coating services were provided by CAM Bioceramics B.V. (Leiden, The Netherlands), a contract manufacturer of calcium phosphate materials. The same coupons for making Ti6Al4V-PMMA constructs were used for coating (Fig 5). Prior to coating, the coupons were sandblasted to enhance coating adhesion. Coatings of thickness 50±5μm were then sprayed onto the substrate using the plasma spray method. The coating thickness was verified using an eddy current device (Fischer® Fischerscope® MMS®). The HA coatings were characterised by Scanning Electron Microscopy (SEM) using a Hitachi TM3000 at a magnification of 100x and 300x (Fig 6). More extensive characterisation of the HA coated coupons (X-ray diffraction analysis, adhesion strength) was performed in an earlier study [24].

**2.1.3. Final sample preparation.** To prevent inconsistent readings of the thermal camera due to differences in thermal emissivity of the different materials (PMMA, HA, Ti6Al4V), all constructs were spray-painted black prior to the experiment (Fig 7). Without the spray-paint, differences in emissivity between biomaterials will make it hard to interpret the resulting thermographs [15]. The paint used was SPECTRUM matte black spray paint.

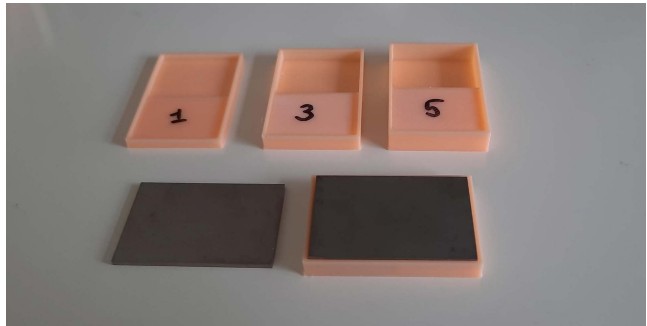

**Fig 3. 3D-printed cement templates of thickness 1, 3 and 5 mm, and Ti6Al4V coupons used to create Ti6Al4V-PMMA constructs with controlled PMMA layer thickness.**

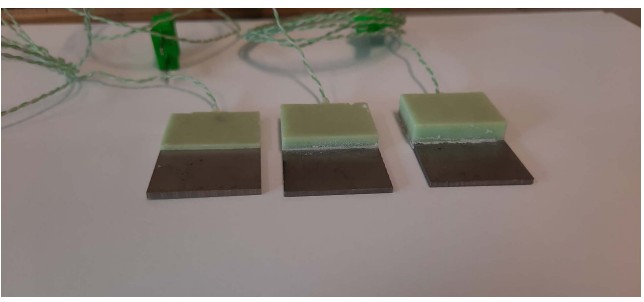

**Fig 4. Ti6Al44-PMMA constructs with a PMMA layer of 1, 3 and 5 mm thickness.**

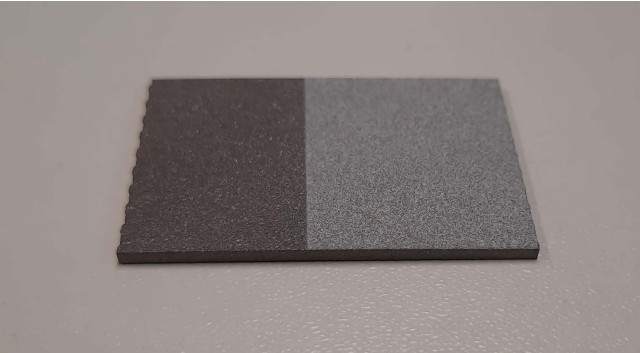

**Fig 5. Ti6Al4V coupon coated with HA.**

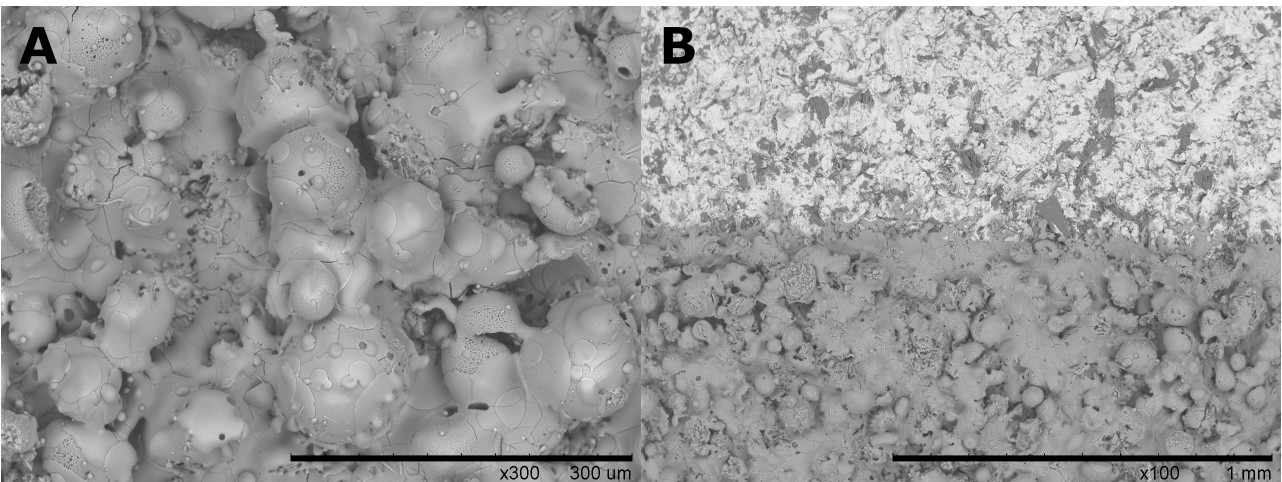

**Fig 6. SEM images of the HA coated coupon made at 100x magnification: A) SEM close-up of the HA coating surface, and B SEM image of the border of the HA coating, with non-coated Ti6Al4V at the top and the HA coating at the bottom.**

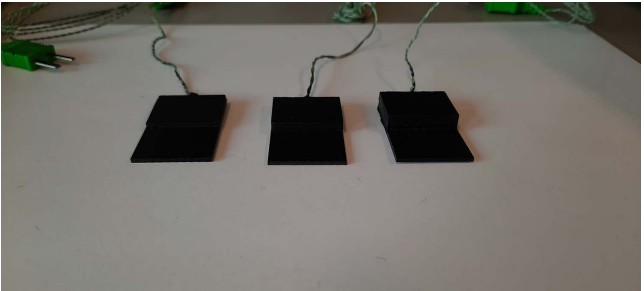

**Fig 7. Ti6Al44-PMMA constructs with a PMMA layer of 1, 3 and 5 mm thickness after spray-painting with black paint.**

**2.1.4. Preparation of PAA gel phantoms.** A PAA gel tissue phantom was used to estimate the magnitude of the heat sink effect of soft tissue surrounding the implant, i.e. the dissipation of heat to surrounding tissues away from the implant-tissue interface. This tissue phantom was produced a according to the method described in ASTM F2182-19 [27]. The PAA gel is designed to mimic the thermal and electrical properties of human tissue [28]. In brief, the gel was prepared by dissolving 10g/L partial sodium salt of polyacrylic acid in saline (1.32g/L NaCl in deionised water). After adding the PAA, the mixture was mixed thoroughly and left to stand for 24h before being used. PAA partial sodium salt and sodium chloride were obtained from Sigma-Aldrich. A second 3D printed template was used to create layers of gel of 5mm thickness. The 3D model files (.STL and.BLEND) used to make an even layer of gel are available at https://doi.org/10.7910/DVN/OZ0HDO. The template was fitted over the construct and filled with gel. After removing excess gel, the templates were removed prior to heating (Fig 8).

## 2.2. Induction heating experiments

**2.2.1. Thermal imaging during induction heating.** The thermal conductivity of HA coatings and PMMA layers was assessed qualitatively using an infrared-spectrum camera (Testo 872 Thermal imageer; Testo SE & Co KGaA, Titisee-Neustadt, Germany). Constructs were heated using an induction heater featuring a pancake-type coil of nine turns of copper litz wire with an inductance of 12 mH, which was also used in an earlier study [14]. Heating was carried out continuously for two minutes, followed by a three-minute cooling period. An infrared-spectrum image was taken every ten seconds throughout the experiment. The thermal imaging experiment was carried out with and without gel phantoms.

**2.2.2. Quantitative temperature measurement.** Quantitative temperature measurements were performed using K-type thermocouples and recorded in PicoLog (version 6.2.12) using a TC-08 Thermocouple data logger (Pico Technology, St. Neots, UK). In addition to the thermocouple beneath the cement layer of the Ti6Al4V-PMMA constructs, two more thermocouples were taped in place on the constructs' top surfaces (Fig 9). Direct induction heating of the thermocouples inside the electromagnetic field was found to be negligible.

Three target temperatures were tested using the induction heater described above: 50, 70 and 90°C. The temperatures 50°C and 70°C represent mild and ablative NCIH treatment regimes, respectively [15]. 90°C is an extreme temperature that is higher than the intended temperature for NCIH treatment of PJI. This temperature represents accidental overheating, or deliberate heating for implant removal [29].

The experiment was started at a temperature of <30°C, consistent with an exposed knee joint that is washed with saline during a DAIR procedure [30]. The alternating magnetic field was applied continuously until $T_{bottom}$ (for PMMA

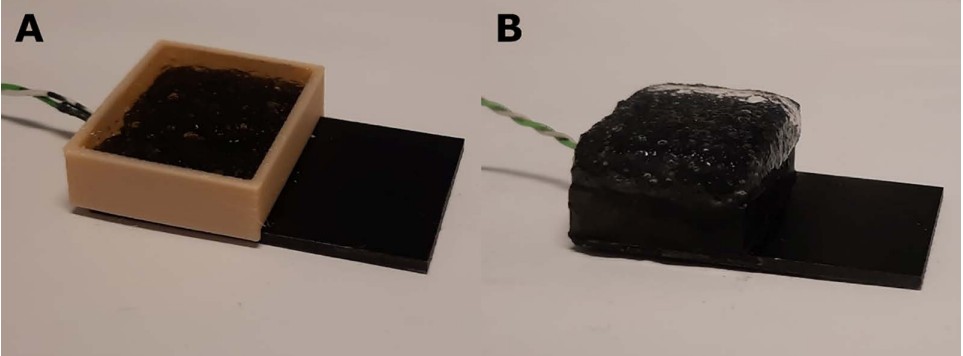

**Fig 8. Addition of the PAA gel to the constructs using a 3D-printed PLA template: A) PLA template filled with PAA gel; B) final construct after removing the template.**

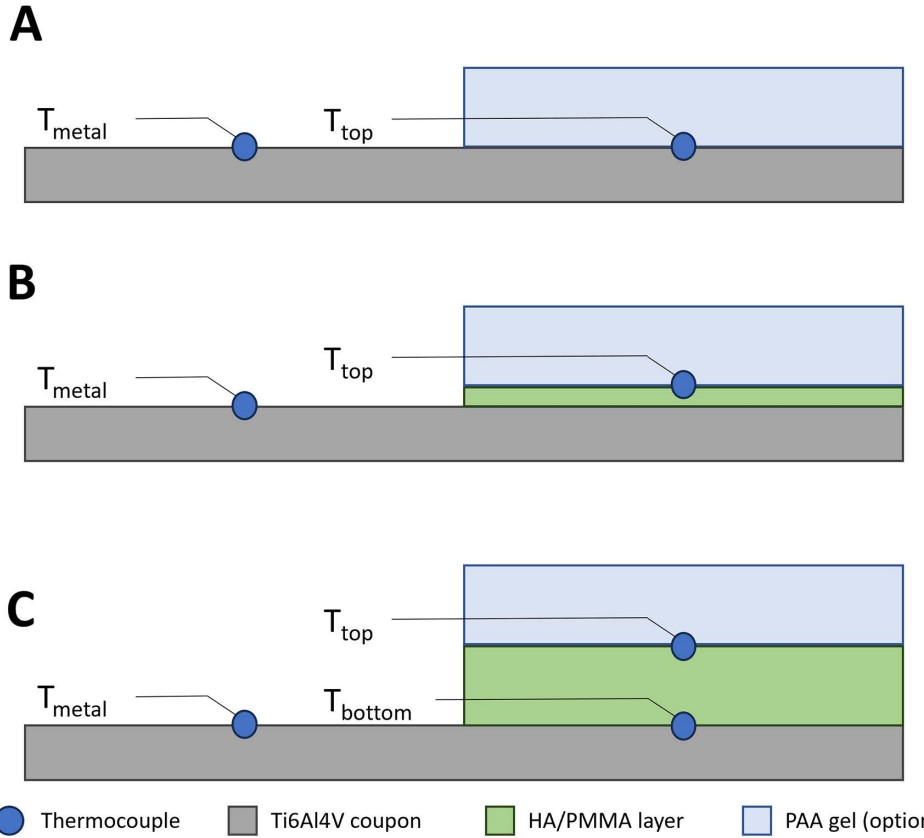

**Fig 9. Diagram of the placement of thermocouples on the different construct modalities: A) negative control/uncemented implants, B) HA-coated implants, and C) cemented implants. For each configuration, Ttop represents the implant-tissue interface.**

constructs) or $T_{metal}$ (for HA coatings and controls) reached the target temperature. After heating stopped, the temperature was recorded until all thermocouples had cooled to <35°C. Data was exported to CSV format and analysed.

Thermal doses were quantified as CEM43 [22,23]. The formula for thermal dose is:

$$CEM43 = \sum_{i=1}^{n} t_i \cdot R^{43-Ttop}$$

Where R is equal to 0 below 37°C, 0.25 between 37°C and 43°C and 0.5 above 43°C. $t_i$ is equal to 1 second (the sampling frequency). A CEM43 was considered to be harmful to surrounding tissues [22]. CEM43 values were calculated in Excel (Microsoft, version 2308). Graphs were made in in RStudio (Posit PBC, version 2022.02.2+485, R version 4.2.0) using the ggplot2 package. All experiments were conducted in duplicate and average values were reported. All experiments were carried out with and without gel phantoms. The total number of experiments across all implant configurations, temperatures and replicates was 60.

## 3. Results

The measured temperature curves for the target temperature of 70°C as well as top-down and cross-sectional thermographs for the negative control, HA coated coupons and coupons with 5 mm PMMA bone cement are shown in Figs 10–12. Thermocouple data and infrared images for all target temperatures and samples are provided in S1 and S2 Figs. When

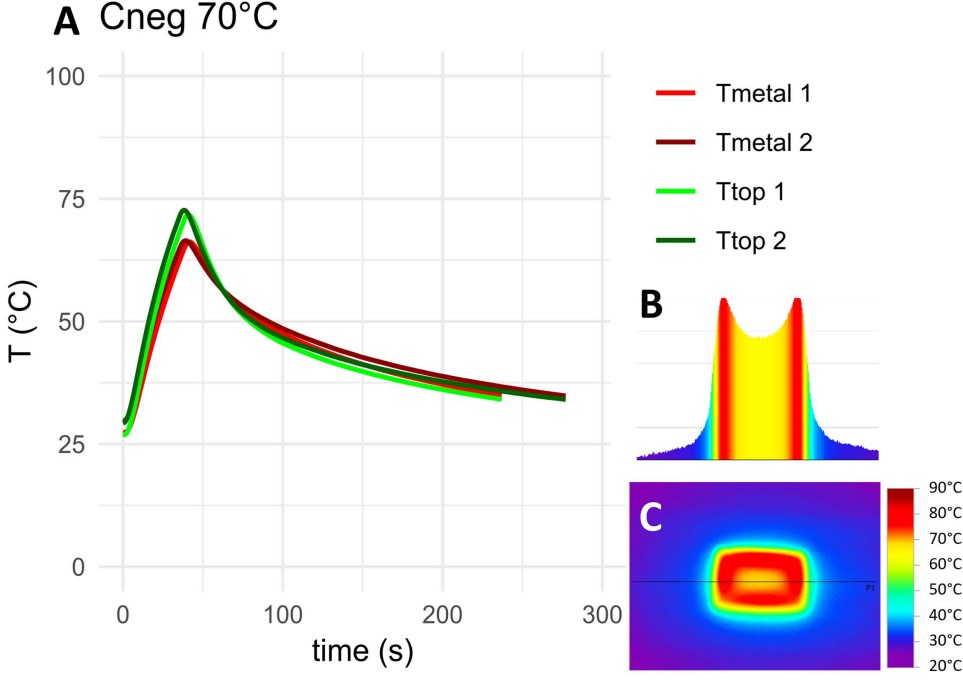

**Fig 10. Thermal measurements on the negative control sample heated to 70°C: A) Temperature curve, B) temperature cross-section and C) thermograph.** The black line in the thermograph indicates the position of the temperature cross-section. Tmetal and Ttop refer to the location of the thermocouple on the construct, see Fig 9. 1 and 2 refer to duplicate measurements.

heated to 50°C, 70°C or 90°C, HA coated coupons exhibited no insulative effects compared to the negative control sample. PMMA bone cement provided thermal insulation depending on the thickness of the cement. With PAA gel phantom, reaching the target temperature for negative controls and HA coated samples took approximately twice as long as without phantom (Figs 13–14 and S1 Figs). This delayed heating was not observed with bone cement-covered samples (Fig 15 and S2 Figs). For all samples, PAA gel phantoms resulted in a reduction of the measured temperature $T_{top}$.

Averaged thermal doses for all samples are given in Fig 16 and Table 1. The starting temperatures, peak temperatures and CEM43 results for individual replicates can be found in S3 Table. The thermal dose ranged from 0 to $9.9*10^{13}$ CEM43. These outcomes were dependent on the heating temperature, the biomaterial and the presence or absence of PAA gel phantoms. Without gel phantom, negative control samples and HA coated coupons could not be heated to 50°C without exceeding the limit value of 16 CEM43. Coupons with cement layers of 1, 3 and 5 mm could be heated to 50°C, 50°C and 70°C, respectively, without exceeding the limit value. When PAA gel phantoms were applied, negative controls and HA coated coupons could be safely heated to 50°C, and the safe temperature for coupons with bone cement increased to 70°C, 90°C and 90°C, respectively.

## 4. Discussion

### 4.1. General findings

This study investigated the safety of induction heating of metal implants with different implant configurations. These biomaterials were selected since they are commonly used clinically in combination with metal implants. Both biomaterials are used to fixate the implant to the bone through different mechanisms. Bone cement provides immediate fixation upon

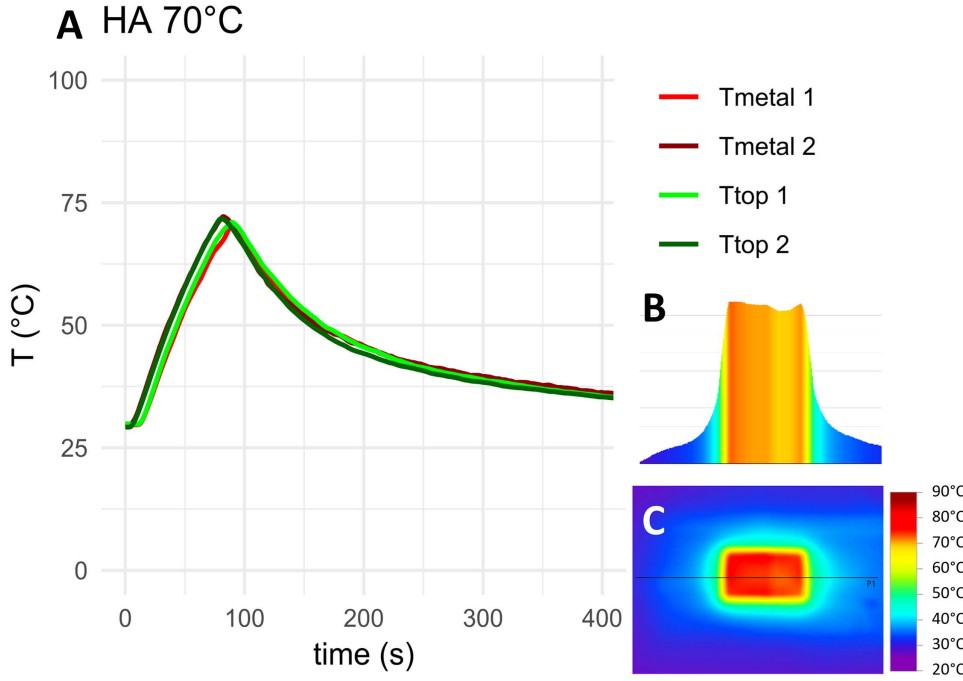

**Fig 11. Thermal measurements on the HA coated sample heated to 70°C: A) Temperature curve, B) temperature cross-section and C) thermograph.** The black line in the thermograph indicates the position of the temperature cross-section. Tmetal and Ttop refer to the location of the thermocouple on the construct, see Fig 9. 1 and 2 refer to duplicate measurements.

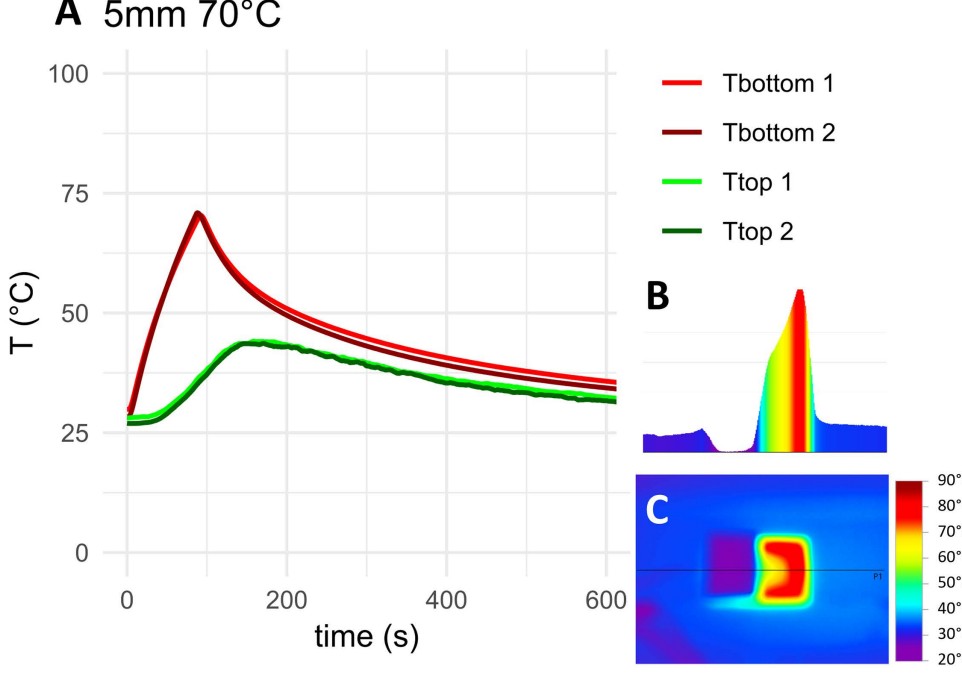

**Fig 12. Thermal measurements on the sample with 5 mm of PMMA bone cement heated to 70°C: A) Temperature curve, B) temperature cross-section and C) thermograph.** The black line in the thermograph indicates the position of the temperature cross-section. Tbottom and Ttop refer to the location of the thermocouple on the construct, see Fig 9. 1 and 2 refer to duplicate measurements.

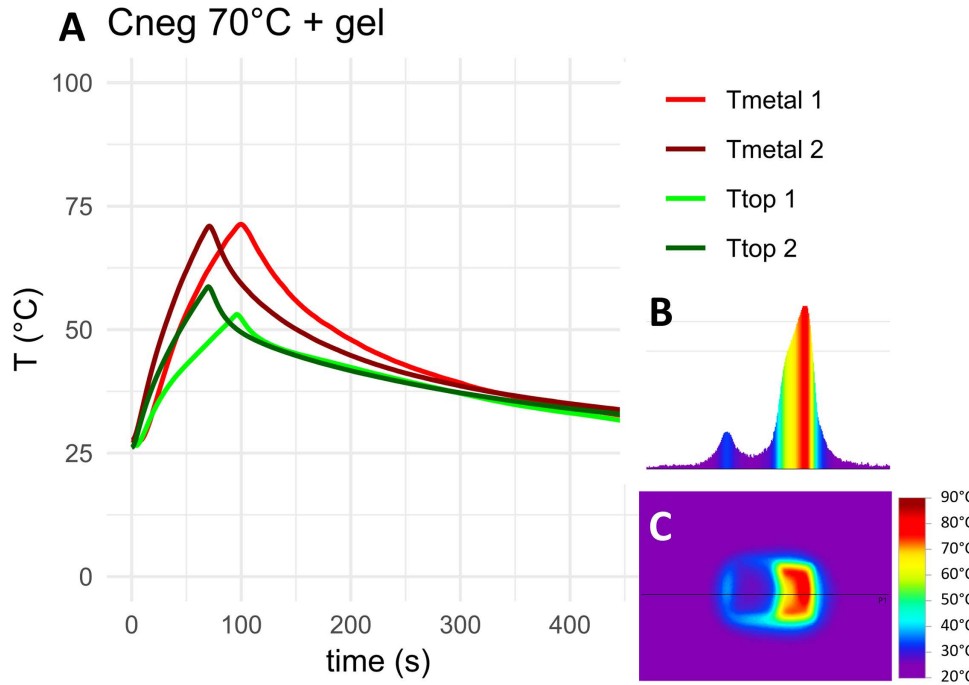

**Fig 13. Thermal measurements on the negative control sample with a PAA gel layer heated to 70°C: A) Temperature curve, B) temperature cross-section and C) thermograph.** The black line in the thermograph indicates the position of the temperature cross-section. Tmetal and Ttop refer to the location of the thermocouple on the construct, see Fig 9. 1 and 2 refer to duplicate measurements.

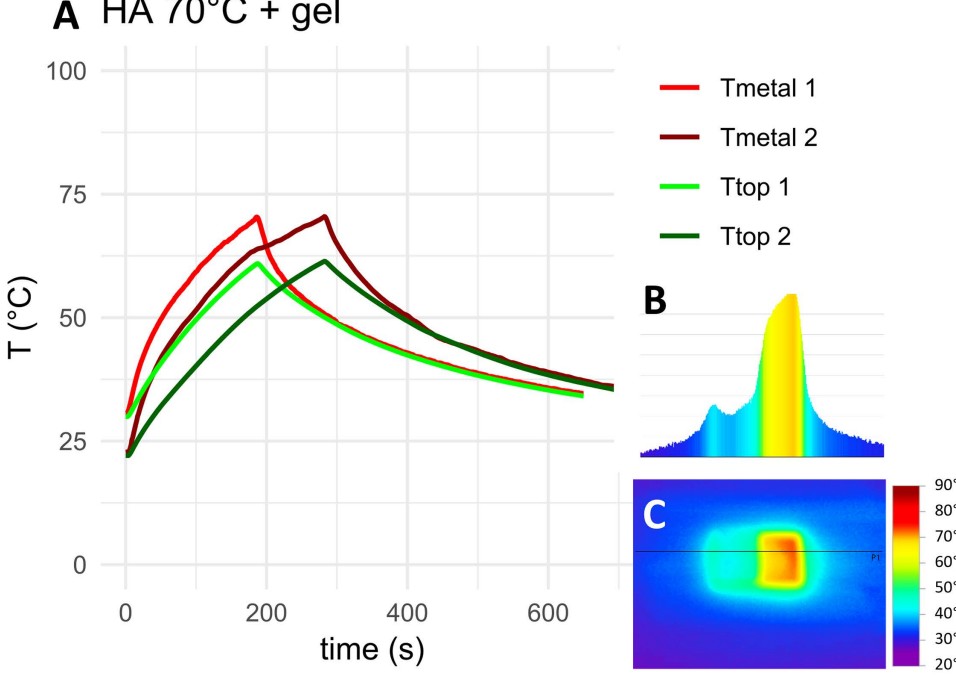

**Fig 14. Thermal measurements on the HA coated sample with a PAA gel layer heated to 70°C: A) Temperature curve, B) temperature cross-section and C) thermograph.** The black line in the thermograph indicates the position of the temperature cross-section. Tmetal and Ttop refer to the location of the thermocouple on the construct, see Fig 9. 1 and 2 refer to duplicate measurements.

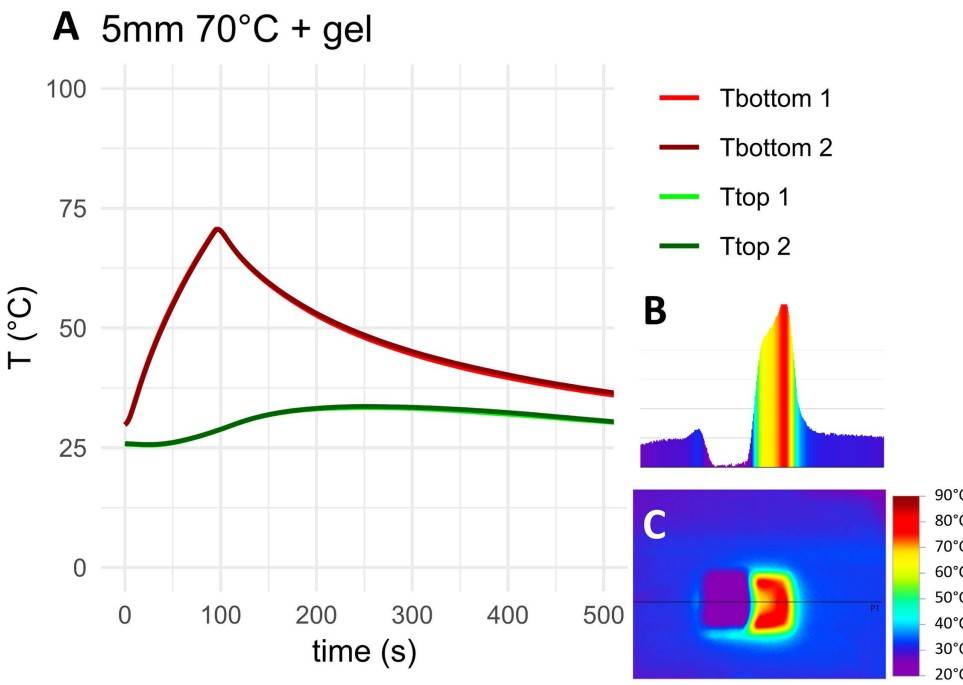

**Fig 15. Thermal measurements on the sample with 5 mm of PMMA bone cement and a PAA gel layer heated to 70°C: A) Temperature curve, B) temperature cross-section and C) thermograph.** The black line in the thermograph indicates the position of the temperature cross-section. Tbottom and Ttop refer to the location of the thermocouple on the construct, see Fig 9. 1 and 2 refer to duplicate measurements.

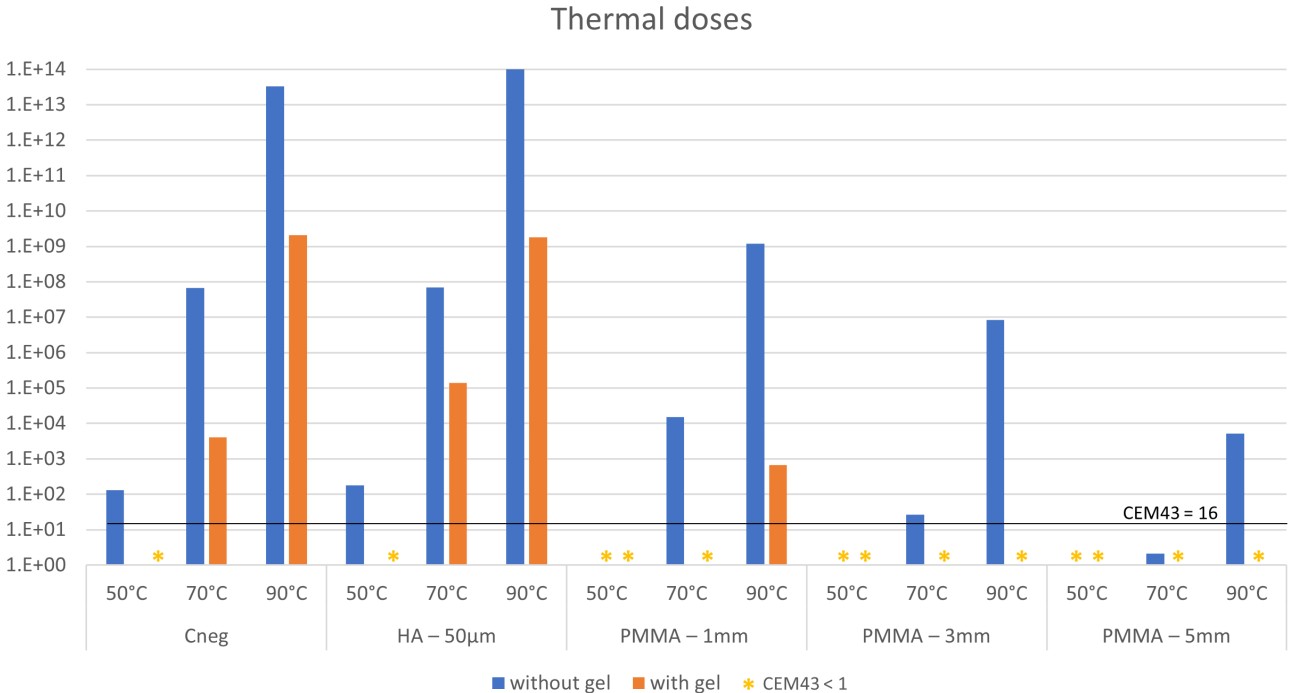

**Fig 16. Thermal doses measured T$_{top}$.**

**Table 1. Tmax and CEM43 values for different samples.** All outcomes represent the average of 2 experiments (the full data for all replicates can be found in S3 Table). Values highlighted in green represent thermal doses below 16 CEM34, considered safe for bone tissue. Ttop refers to the location of the thermocouple on the construct, see Fig 9.

| Sample | Target | No gel | | gel | |
|---|---|---|---|---|---|
| | | $T_{top}$, max | $CEM43_{top}$ | $T_{top}$, max | $CEM43_{top}$ |
| Cneg | 50°C | 53.0 | 1.3E+02 | 44.3 | 9.7E-01 |
| | 70°C | 72.2 | 6.6E+07 | 55.9 | 4.1E+03 |
| | 90°C | 91.1 | 3.3E+13 | 76.4 | 2.1E+09 |
| HA – 50µm | 50°C | 52.6 | 1.8E+02 | 40.2 | 5.3E-03 |
| | 70°C | 71.4 | 7.0E+07 | 61.2 | 1.4E+05 |
| | 90°C | 91.7 | 9.9E+13 | 73.7 | 1.8E+09 |
| PMMA – 1 mm | 50°C | 43.1 | 6.2E-01 | 30.6 | 0.0E+00 |
| | 70°C | 57.2 | 1.5E+04 | 40.3 | 1.3E-02 |
| | 90°C | 73.1 | 1.2E+09 | 52.6 | 6.6E+02 |
| PMMA – 3 mm | 50°C | 36.4 | 0.0E+00 | 28.6 | 0.0E+00 |
| | 70°C | 48.0 | 2.7E+01 | 33.1 | 0.0E+00 |
| | 90°C | 64.4 | 8.3E+06 | 39.7 | 1.6E-02 |
| PMMA – 5 mm | 50°C | 35.1 | 0.0E+00 | 28.0 | 0.0E+00 |
| | 70°C | 43.9 | 2.1E+00 | 33.5 | 0.0E+00 |
| | 90°C | 55.4 | 5.1E+03 | 41.2 | 1.5E-01 |

hardening in situ, while HA coatings facilitate fixation of the implant by promoting bone growth on the implant surface (osteoconduction).

As expected, the results of our study showed an insulating effect for PMMA, which increased with cement thickness. No insulating effect was found for HA coatings. Without a PAA gel layer, the safe temperature for the metal-biomaterial interface was < 50°C for non-coated and HA coated implants, and 50°C, 50°C and 70°C for implants with a cement mantle of 1,3 and 5mm, respectively. These temperatures increased to 50°C for non-coated and HA coated implants and 70°C, 90°C and 90°C for implants with a cement mantle of 1,3 and 5mm, respectively, in the presence of a PAA gel phantom.

**4.1.1. HA coatings and non-coated implants.** Typically, HA coatings have a thickness ranging from 25 to 150µm and are applied to the implant by the manufacturer [21,31]. The most common coating method is plasma spraying, where an HA base powder is melted in a plasma jet and accelerated towards the implant surface. In a previous study, we have shown that HA coatings on grade 5 titanium were stable at temperatures over 100°C. Consequently, there is no risk of damaging the coating during NCIH treatment [24].

In the absence of any insulating effect of the HA-coating, HA-coated implants should be heated to a maximum temperature of 50°C at the implant-coating interface, based on a maximum thermal dose of 16 CEM43. Previously, it was shown that full eradication of an established biofilm was possible by combining antibiotics treatment with NCIH using temperatures of 65°C for 3.5 minutes [14,15,18]. While this temperature cannot be safely reached at the implant-tissue interface, heating to 50°C was still shown to have a synergistic effect with antibiotics treatment, although total eradication is not achieved [14]. Furthermore, higher temperatures can be reached with increasing distance from the implant-coating interface. Indeed, previous research has shown that high thermal doses (in excess of 7000 CEM43) can be achieved by segmental induction heating without exceeding 16 CEM43 at regions critical for fixation due to the heat sink effect of the metal implant and the poor thermal conductivity of titanium alloys [15].

**4.1.2. Cemented implants.** In contrast to HA coatings, PMMA bone cement mantles are typically <1–5mm thick, depending on the type of implant [19,20,26]. Bone cement cures in situ from an unpolymerized paste that is injected into the bone prior to insertion of the implant. Although thicker bone cement is generally more stable, there is an increased

chance of tissue necrosis due to heat release during the exothermic polymerisation reaction, making cement mantles exceeding 5 mm in thickness undesirable [19].

In the presence of a PAA gel layer, we found that cement mantles of 3 mm or thicker could reduce the thermal dose below 16 CEM43 even when the titanium coupon was heated to 90°C. However, heating to such high temperatures is undesirable due to limits in the temperature stability of PMMA bone cement. Previously, it was shown that no structural changes occurred in PMMA bone cement upon heating up to 70°C, whereas heating to 100°C resulted in small changes in the crystallinity of the cement [32]. Furthermore, PMMA is only mechanically stable at temperatures below 80°C [24]. Consequently, for NCIH for cement mantles thicker than 1 mm, loss of mechanical function is a greater risk than thermonecrosis. Therefore, a maximum temperature of 80°C at the implant-biomaterial interface should be considered for NCIH on cemented implants.

Interestingly, heating above 80°C has been suggested as a strategy to facilitate implant removal during revision surgery [33]. Previous research has experimentally and computationally modelled the heat transfer at the metal-cement interface. However, the heat transferred to surrounding tissue has not been investigated. Our data suggests that this method might be viable for cement mantles >3 mm thick, although it is outside the scope of this investigation.

### 4.2. Limitations & strengths

There are some areas where our model may differ from the clinical situation, potentially leading to differences in the CEM43 values. Firstly, there was some variation in the temperature measured by one thermocouple between experiments as well as the temperature measured between different thermocouples, even in negative controls. There are two possible causes for these variations:

1. Variations between replicates may be attributed to asymmetrical placement of the coupons within the magnetic field. If a construct is placed less optimally in the magnetic field of the induction heating device, more time is required to reach the target temperature. Variation in starting temperature (due to variations in ambient temperature) also led to differences in the temperature measured between duplicate experiments. For all experiments, both replicates fell either below or above the safe limit of CEM43 = 16. Therefore, any variation in the measured CEM34 values did not affect the outcome of the test.

2. Variations between the measured temperature of the two halves of the negative controls may be attributed to asymmetrical placement of the thermocouples. As can be seen on thermography images, coupons heated up more quickly on the edges compared to the centre of the coupon. Consequently, small variations in the placement of the thermocouples could be the cause of the variation in measured temperatures.

Overall, we do not expect the outcomes of our investigation to be affected by these variations, as their magnitude was small compared to the differences between the different study groups.

Secondly, the constructs used in this study are a simplification of real implants and do not capture some important aspects:

1. The time required for heating could be different for real implants, due to their greater mass and complex geometry compared to our model. This could increase the thermal dose measured on the implant-tissue interface. However, a previous study showed that similar timespans were needed to heat real implants to 60°C (10-60s) compared to our model (20–70 seconds). Therefore, it is unlikely that the difference in implant size leads to differences in thermal dose delivered to surrounding tissue.

2. The PAA gel phantoms used are static and have a limited volume. They therefore do not account for cooling mechanisms present in living tissue (e.g., blood flow). Consequently, it is expected that the phantoms provide an overly conservative estimate of the cooling provided by real tissues [28,34]. On the other hand, the phantoms are able to dissipate some heat through moisture evaporation, which is not the case for deep tissues in humans.

3. The duration of heating might differ between a clinical setting and our model. In this study, heating was immediately stopped once the target temperature was reached. However, different heating strategies could be employed in the clinic, such as sustained heating over a longer period of time or pulsed [14–16]. These alternative strategies could increase the final thermal dose that is delivered to surrounding tissues.

In summary, there are some limitations to our model that could influence the real thermal dose delivered to surrounding tissue during NCIH. Despite these limitations, modelling is an essential part of introducing new technologies to be used in humans, to provide optimal safety in the clinic. Our model provides very relevant insights into the capacity for thermal shielding of two biomaterials (HA coatings and PMMA bone cement) that are commonly found in combination with metal total joint replacement implants. Thus, it allows us to estimate the range of temperatures that can be reached safely without thermal damage to surrounding tissues. This is an essential step towards clinical acceptance and application of induction heating as a treatment option for PJI.

### 4.3. Outlook

Before NCIH can be implemented in the clinic, several aspects require further investigation. Primarily, a method for controlling the temperature at the implant-tissue interface during NCIH treatment is needed For invasive NCIH treatment, such as DAIR, thermocouples or fibre-optic sensors can be used. As an alternative for non-invasive NCIH treatment, acoustic sensing has been suggested to detect boiling at the implant-tissue interface in order to control the temperature [35]. However, this method results in temperatures of 80 degrees C and higher and cannot be used for lower temperatures. Hence, more research is needed to develop non-invasive temperature control systems. Furthermore, induction devices must be specifically designed with invasive or non-invasive treatment in mind. While devices used for invasive treatment should be relatively small and able to reach the implant around the tissues that are present, a device used for non-invasive treatment should be able fit around the affected limb, or remain to one side. Additionally, non-invasive devices have to be adjustable for different patient body types or BMI.

An alternative avenue of investigation is to use NCIH for prevention of PJI, rather than treatment of existing infections. In this scenario, a potential strategy could be to maintain elevated physiological temperatures (between 37°C and 46°C) for a period of minutes or hours, using non-invasive induction heating as a heat source. As this strategy is potentially less impactful than heating to high temperatures, it is interesting to consider for infection prevention, rather than treatment of existing infections. This strategy mimics fever conditions generated by the body after infection. Previous research suggests that elevated temperatures stimulate the native immune system in addition to stunting bacterial growth [36,37]. The effects on bacteria and human tissues under these conditions are currently being investigated. Another key point of consideration is to create protocols to make the decision to start preventative intervention. This decision can be based on patient-related (i.e. diabetes) or post-operative (i.e. persistent wound leakage) risk factors.

### 5. Conclusion

This study aimed to investigate the feasibility of NCIH treatment for common implant configurations. Our findings suggest that non-coated implants and HA-coated implants can be heated to 50°C at the implant-coating interface without thermal damage to surrounding tissues. For cemented implants, this temperature threshold increases to 70–90°C, depending on the thickness of the cement mantle.

### Supporting information

**S1 Figs. Thermocouple data.**
(DOCX)

**S2 Figs. Thermographs.**
(ZIP)

**S3 Table. CEM43 data for individual replicates.**
(XLSX)

## Author contributions

**Conceptualization:** Robert Kamphof, Rob G.H.H. Nelissen, Bart G.C.W. Pijls.

**Data curation:** Robert Kamphof.

**Formal analysis:** Robert Kamphof.

**Funding acquisition:** Rob G.H.H. Nelissen, Giuseppe Cama.

**Investigation:** Robert Kamphof.

**Methodology:** Robert Kamphof, Bart G.C.W. Pijls.

**Project administration:** Rob G.H.H. Nelissen, Giuseppe Cama, Bart G.C.W. Pijls.

**Resources:** Giuseppe Cama, Bart G.C.W. Pijls.

**Supervision:** Rob G.H.H. Nelissen, Bart G.C.W. Pijls.

**Validation:** Robert Kamphof, Bart G.C.W. Pijls.

**Visualization:** Robert Kamphof.

**Writing – original draft:** Robert Kamphof.

**Writing – review & editing:** Robert Kamphof, Rob G.H.H. Nelissen, Giuseppe Cama, Bart G.C.W. Pijls.

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
