## [Decision Letter · Decision Letter 0]

29 Sep 2025

Dear Dr. Kamphof,

Thank you for submitting your manuscript to PLOS ONE. After careful consideration, we feel that it has merit but does not fully meet PLOS ONE’s publication criteria as it currently stands. Therefore, we invite you to submit a revised version of the manuscript that addresses the points raised during the review process.

We look forward to receiving your revised manuscript.

Kind regards,

Abbas Farmany

Academic Editor

PLOS ONE

“This publication is part of the project DARTBAC (with project number NWA.1292.19.354) of the research programme NWA-ORC which is (partly) financed by the Dutch Research Council (NWO). B.G.C.W. Pijls received funding from ZonMW under Veni Grant 09150161810084.”

4. Thank you for stating the following in the Funding Section of your manuscript:

“This publication is part of the project DARTBAC (with project number NWA.1292.19.354) of the research programme NWA-ORC which is (partly) financed by the Dutch Research Council (NWO). B.G.C.W. Pijls received funding from ZonMW under Veni Grant 09150161810084.

R.G.H.H. Nelissen and B.G.C.W. Pijls are listed as inventors on patents from Leiden University Medical Center regarding induction heating of metal implants.

This article was written in collaboration with CAM Bioceramics B.V., situated in Leiden, The Netherlands. CAM Bioceramics B.V. is a contract development and manufacturing organisation that designs, develops and manufactures calcium phosphates for a range of medical applications.”

“This publication is part of the project DARTBAC (with project number NWA.1292.19.354) of the research programme NWA-ORC which is (partly) financed by the Dutch Research Council (NWO). B.G.C.W. Pijls received funding from ZonMW under Veni Grant 09150161810084.”

“I have read the journal's policy and the authors of this manuscript have the following competing interests:

R.G.H.H. Nelissen and B.G.C.W. Pijls are listed as inventors on patents from Leiden University Medical Center regarding induction heating of metal implants.

This article was written in collaboration with CAM Bioceramics B.V., situated in Leiden, The Netherlands. CAM Bioceramics B.V. is a contract development and manufacturing organisation that designs, develops and manufactures calcium phosphates for a range of medical applications.”

Reviewers' comments:

Reviewer's Responses to Questions

**Comments to the Author**

1. Is the manuscript technically sound, and do the data support the conclusions?

Reviewer #1: Yes

Reviewer #2: Partly

2. Has the statistical analysis been performed appropriately and rigorously?

Reviewer #1: No

Reviewer #2: N/A

3. Have the authors made all data underlying the findings in their manuscript fully available?

Reviewer #1: No

Reviewer #2: No

4. Is the manuscript presented in an intelligible fashion and written in standard English?

Reviewer #1: Yes

Reviewer #2: Yes

Reviewer #1: The article presents experimental research on the effect of polymethyl methacrylate (PMMA) and hydroxyapatite (HA) coatings on thermal conductivity in the context of non-contact inductive heating (NCIH) of metallic orthopedic implants. The aim of the study was to determine safe temperature ranges for different implant configurations in order to minimize the risk of tissue damage during the treatment of periprosthetic joint infections (PJI). The experiments were conducted using Ti6Al4V titanium coupons, PMMA layers of varying thickness, and HA coatings, as well as polyacrylamide (PAA) gel phantoms simulating soft tissues.

The article is well-structured, containing an extensive introduction that addresses clinically relevant issues (PJI, biofilm, antibiotic resistance) and provides a detailed description of the methodology. The results are presented clearly, supported by rich graphical documentation and quantitative analysis (CEM43). The study has significant practical relevance in orthopedics and biomaterials engineering.

In my opinion, the manuscript is valuable; however, before further processing, it will require several revisions and additions. Detailed comments are provided below.

Major comments:

The introduction only briefly addresses, and in some aspects overlooks, several critical factors influencing the mechanical properties of bone cements. These include the specific characteristics of the mixing process, the presence of contaminants that can naturally occur in the surgical field—such as blood, bone debris, or saline solutions—as well as even minor deviations in the proportions of components in binary materials. Each of these factors can substantially alter the cement’s microstructure, polymerization kinetics, and long-term performance in vivo, ultimately affecting its clinical effectiveness. A more comprehensive discussion of these aspects is warranted, supported by relevant literature highlighting the effects of mixing protocols, intraoperative contamination, and compositional imbalances on the mechanical integrity and reliability of bone cements (DOI: 10.3390/ma12233963; 10.3390/ma12193073; 10.3390/ma15165577; 10.3390/ma15062197).

The Ti6Al4V–PMMA and Ti6Al4V–HA constructs used in this study have simplified geometry and reduced mass compared to actual implants, which may influence the real temperature distribution. In future work, it is recommended to extend the study to models with geometries more closely resembling clinical implants. For the current manuscript, the discussion should be supplemented with literature data comparing laboratory conditions with in vivo scenarios.

The PAA gel phantoms do not account for dynamic cooling by blood flow and physiological fluids. The authors are encouraged to include numerical simulations or literature-based models addressing the perfusion effect in tissues to better estimate the safety of NCIH under clinical conditions.

The results are presented mainly as averages from two measurements, without an analysis of statistical significance. It would be advisable to perform statistical analysis (e.g., ANOVA or non-parametric tests) to assess differences between sample configurations and temperatures; if this is not feasible, the limitation should be explicitly described in the manuscript.

Although threshold temperatures for PMMA and HA are mentioned, the study does not include post-heating assessments of, for example, changes in mechanical properties or coating adhesion. Future studies should incorporate mechanical and microscopic testing after heating cycles.

The discussion of biological responses, including osteointegration and the effects of cement on surrounding bone tissue, requires further development. This should be complemented by an examination of factors—such as the incorporation of solid admixtures (e.g., ceramics, glassy carbon)—that have the potential to enhance both the mechanical performance and osteointegration of bone cements. Such modifications can influence not only the cement’s load-bearing capacity but also its bioactivity, promoting stronger and more durable bonding at the bone–cement interface. Detailed insights into these effects can be found in the series of studies Effect of Various Admixtures on Selected Mechanical Properties of Medium Viscosity Bone Cements: Part 1, Part 2, and Part 3, which should be reviewed and appropriately cited to complete the reference list.

Although synergy with antibiotics is mentioned, the manuscript lacks data on the effectiveness of NCIH in reducing biofilm at different temperatures. The authors are encouraged to expand the discussion to address the mechanisms by which sub-thermal heat doses affect biofilm and the potential implications for antibiotic resistance.

The figures and tables are numerous but not always easy to interpret (e.g., small axis labels, lack of standardized color schemes). It would be beneficial to standardize figure formatting, increase the readability of labels, and apply a consistent color scheme throughout the manuscript.

Reviewer #2: The authors submitted an original research article entitled “Thermal insulation of poly(methyl methacrylate) bone cement and hydroxyapatite coatings under induction heating of metal implants”.

Background of the study was to use induction heating of metal implants to supplement existing therapy options for implant infections like DAIR (debridement, antibiotics and implant retention).

Therefore, titanium coupons were heated to different temperatures and heat transfer through different materials were measured.

The topic of the paper is important in the field of orthopedics and PJI and of interest to the readers of PLOS one. Overall, the paper is well structured and written. The set up and the results are presented clearly. However, there are some major points which have to be clarified: Firstly, the pictured application is difficult to envision. The schematic of figure 1 is somehow idealized, the situation in the operating theatre is heterogenous and an exposition of the whole transition from the femoral head to the stem is difficult due to the massive musculature which is present. It is not clear how an induction coil would be placed to reach the desired parts of the implant successfully and adequately. Furthermore, it seems difficult to understand that the overlying tissue layer – here in the model represented by the PAA gel – is called “heat sink”. The layer which acts in this set up as heat sink is the actual tissue which will be damaged by absorbing parts of the heat. The temperature range (< 30°C) given at which the experiments were conducted is rather broad. Since the surrounding temperature is a crucial factor for the experiments so the exact temperature for each experimental run should be stated to allow for a correct evaluation. Lastly, it is not clear why the measurements of each group are only performed twice. While in some groups the double measurement led to relatively similar values, there were more obvious differences in others. Although the presence of these variation in the values are discussed in the limitations & strengths section, the actual values of the single measurements are nowhere stated. They must be included in the paper. Repetition of the measurement will clarify the reason for the variation and if the reason given in the limitations & strengths section is correct. It is strongly suggested to increase the number of measurements or give a reasonable explanation for not doing so.

Furthermore, the following points should be revised:

Section 2.1.3 Final sample preparation: Please include some short but clear information about the used spray-paint. The given reference (15) is a paper which also gives no more detailed information but instead also references another literature which is not freely available.

Section 2.2.2 Quantitative measurement: The exact total number of the experiments for each group has to be included.

Fig. 10 – 15: In the Figure captions 90°C are stated while in the figures themselves 70°C is stated. Please revise correctly. Additionally, the thermographic pictures have to be explained better, maybe by adding one schematic, especially for the upper part: what means the different height of the coloured graph? For readers not familiar with this technique this is not self-explainable.

Table 1: For Ttop max the standard deviation should be stated. Furthermore, it is confusing that for the control implants (Cneg) also the term “T top” is used. Shouldn´t it be “T bottom”? Please clarify and consider adding a respective foot note to the table.

Outlook: While alternative strategies for preventing bacterial growth are of great importance, the mentioned strategy here has to be explained further. How should the coil be applied in a closed-skin/prothesis-in-situ situation be placed? What time-points should be chosen for heating? How do you want to decide when to heat the implant for prevention purposes?

**Do you want your identity to be public for this peer review?** For information about this choice, including consent withdrawal, please see our Privacy Policy

Reviewer #1: No

Reviewer #2: No

---

## [Author Response · Author response to Decision Letter 1]

24 Oct 2025

Journal Requirements

Response:

Thank you for letting us know where our manuscript deviated from the Plos One requirements. We have amended the sections indicated by you; please let us know if any additional changes are necessary.

Reviewer 1

Comment 1:

The introduction only briefly addresses, and in some aspects overlooks, several critical factors influencing the mechanical properties of bone cements. These include the specific characteristics of the mixing process, the presence of contaminants that can naturally occur in the surgical field—such as blood, bone debris, or saline solutions—as well as even minor deviations in the proportions of components in binary materials. Each of these factors can substantially alter the cement’s microstructure, polymerization kinetics, and long-term performance in vivo, ultimately affecting its clinical effectiveness. A more comprehensive discussion of these aspects is warranted, supported by relevant literature highlighting the effects of mixing protocols, intraoperative contamination, and compositional imbalances on the mechanical integrity and reliability of bone cements (DOI: 10.3390/ma12233963; 10.3390/ma12193073; 10.3390/ma15165577; 10.3390/ma15062197).

Response 1:

Dear reviewer, thank you for your feedback and for providing these interesting references for us to read. We would like to point out that our study concerns the thermal properties of PMMA bone cement and HA coatings in the context of non-contact induction heating (NCIH). The mechanical properties of the cement pre and post NCIH are not within the scope of the current investigation, but have been studied in the past by us (https://doi.org/10.12688/f1000research.148225.2).

Comment 2:

The Ti6Al4V–PMMA and Ti6Al4V–HA constructs used in this study have simplified geometry and reduced mass compared to actual implants, which may influence the real temperature distribution. In future work, it is recommended to extend the study to models with geometries more closely resembling clinical implants. For the current manuscript, the discussion should be supplemented with literature data comparing laboratory conditions with in vivo scenarios.

Response 2:

Thank you for pointing this out. We have already addressed the expected impact of different implant size and geometry compared to our model in the “Limitations & strengths” section of the manuscript. We have copied the relevant section here for your convenience:

“The time required for heating could be different for real implants, due to their greater mass and complex geometry compared to our model. This could increase the thermal dose measured on the implant-tissue interface. However, a previous study showed that similar timespans were needed to heat real implants to 60°C (10-60s) compared to our model (20-70 seconds). Therefore, it is unlikely that the difference in implant size leads to differences in thermal dose delivered to surrounding tissue.”

Comment 3:

The PAA gel phantoms do not account for dynamic cooling by blood flow and physiological fluids. The authors are encouraged to include numerical simulations or literature-based models addressing the perfusion effect in tissues to better estimate the safety of NCIH under clinical conditions.

Response 3:

Thank you for pointing this out. We have already addressed the expected differences between PAA gel and living tissues in the “Limitations & strengths” section of the manuscript. The gel phantoms used in our study are created using a procedure from the international standard ASTM F2182-19. The gel phantoms created using this procedure have a thermal capacity and thermal conductivity constant similar to human tissues.

Nevertheless, there is reason to believe that living tissue will have increased heat sink capacity compared to the gel phantoms used in this study (see studies that modelled the effect of blood flow in hyperthermia situations: https://doi.org/10.1016/0360-3016(79)90725-9 and https://doi.org/10.1088/2057-1976/ad0398). As a consequence, our model can be seen as a ‘worst case scenario’, and the real situation will lead to an even greater cooling effect. Since the possibility of using NCIH has been demonstrated even under these unfavourable conditions, we would argue that it is unnecessary to perform even more in vitro studies, and that the next step would be to test NCIH in ex vivo or in vivo scenarios.

Comment 4:

The results are presented mainly as averages from two measurements, without an analysis of statistical significance. It would be advisable to perform statistical analysis (e.g., ANOVA or non-parametric tests) to assess differences between sample configurations and temperatures; if this is not feasible, the limitation should be explicitly described in the manuscript.

Response 4:

Thank you for your feedback. We are of the opinion that statistical analysis of data based on two replicate measurements will not reveal any insights not apparent from qualitative consideration of the data. Our research aim was to establish the relationship between implant modality and the potential temperatures that could be safely employed during NCIH treatment. Since statistical significance does not imply clinical significance and a clear trend in the thermal doses received at the implant surface based on implant modality could be seen, we were satisfied with our current data. There is a growing resistance against the (wrongful) use of p-values (e.g. “Scientists rise up against statistical significance”, https://doi.org/10.1038/d41586-019-00857-9) and statistical significance does not equate to clinical significance. Achieving a statistically significant estimation of the exact temperature threshold where the CEM43 of our model exceeds 16 is not clinically relevant. Finally, we would like to point out that the CEM43 threshold is exceeded by multiple orders of magnitude in implant modalities considered ‘unsafe’ in our study. Repeating the experiments simply to be able to perform statistical tests on the data will not lead to better clinical outcome for patients.

Comment 5:

Although threshold temperatures for PMMA and HA are mentioned, the study does not include post-heating assessments of, for example, changes in mechanical properties or coating adhesion. Future studies should incorporate mechanical and microscopic testing after heating cycles.

Response 5:

We have already performed a study on the mechanical strength of these materials post induction heating. Our publication is available in open access at https://doi.org/10.12688/f1000research.148225.2.

Comment 6:

The discussion of biological responses, including osteointegration and the effects of cement on surrounding bone tissue, requires further development. This should be complemented by an examination of factors—such as the incorporation of solid admixtures (e.g., ceramics, glassy carbon)—that have the potential to enhance both the mechanical performance and osteointegration of bone cements. Such modifications can influence not only the cement’s load-bearing capacity but also its bioactivity, promoting stronger and more durable bonding at the bone–cement interface. Detailed insights into these effects can be found in the series of studies Effect of Various Admixtures on Selected Mechanical Properties of Medium Viscosity Bone Cements: Part 1, Part 2, and Part 3, which should be reviewed and appropriately cited to complete the reference list.

Response 6:

Dear reviewer, thank you for pointing out these interesting aspects of bone cement chemistry. As we pointed out in Response 1, our study focused on studying the thermal properties of implant modalities that are currently commonly used in the clinic, in the context of infection treatment using NCIH, rather than mechanical or biological properties of these materials.

Comment 7:

Although synergy with antibiotics is mentioned, the manuscript lacks data on the effectiveness of NCIH in reducing biofilm at different temperatures. The authors are encouraged to expand the discussion to address the mechanisms by which sub-thermal heat doses affect biofilm and the potential implications for antibiotic resistance.

Response 7:

The effect of NCIH on biofilms in vitro as well as the synergistic effect of NCIH and antibiotics has already been investigated in depth in several studies, please see the following references:

• https://doi.org/10.1302/2046-3758.119.BJR-2022-0010.R1

• https://doi.org/10.1186/s13018-024-04785-x

• https://doi.org/10.1302/2046-3758.145.BJR-2024-0341.R1

• https://doi.org/10.1302/2046-3758.94.BJR-2019-0274.R1

Comment 8:

The figures and tables are numerous but not always easy to interpret (e.g., small axis labels, lack of standardized color schemes). It would be beneficial to standardize figure formatting, increase the readability of labels, and apply a consistent color scheme throughout the manuscript.

Response 8:

Thank you for your comment. Our colour scheme is as follows: for diagrams of our experimental setup, we use grey for metal parts, green for PMMA bone cement or HA coatings, light blue for PAA gel and beige for bone tissue. For temperature measurements with thermocouple, we use green for the top surface of the PMMA or HA, and red for the control side. For CEM43 values in Figure 16 specifically, we use blue and orange to specify the groups with and without gel, specifically. Please, let us know which specific changes we can make to improve our figures. 

Reviewer 2

Comment 1:

Firstly, the pictured application is difficult to envision. The schematic of figure 1 is somehow idealized, the situation in the operating theatre is heterogenous and an exposition of the whole transition from the femoral head to the stem is difficult due to the massive musculature which is present. It is not clear how an induction coil would be placed to reach the desired parts of the implant successfully and adequately.

Response 1:

Dear reviewer, thank you for your comments. Figure 1 of our manuscript is indeed idealised for clarity. In particular, the induction coil pictured in Figure 1 is deliberately oversized for clarity. In reality, an induction device designed for this purpose would be much smaller and designed to fit over the femoral head. Please see this publication, which features a device in Fig. 5 with a similar size and appearance: https://doi.org/10.1302/2046-3758.121.BJR-2022-0216.R1

We hope we have satisfied your curiosity with this answer. Please let us know if we can provide additional information.

Comment 2:

Furthermore, it seems difficult to understand that the overlying tissue layer – here in the model represented by the PAA gel – is called “heat sink”. The layer which acts in this set up as heat sink is the actual tissue which will be damaged by absorbing parts of the heat.

Response 2:

Thank you for pointing out this apparent contradiction. Implying the function of the gel is simply a heat sink was not our intention; naturally, the gel itself represents the very tissue that must be protected from thermal damage. To avoid confusion, we have removed most references of the words ‘heat sink’ from the manuscript, except where it is specifically relevant to discuss this property.

We measured the thermal doses at the implant-tissue interface (i.e. Ttop), which is the site of the highest thermal dose administered to living tissue. The gel phantom, in this scenario, represents the surrounding tissue and can indeed act as a heat sink that decreases the temperature of the implant-tissue interface. Since the temperature of the deeper tissue will never be higher than at interface (as evidenced by the thermographs in S2), we can be sure that if the CEM43 at the interface is below the safe upper limit, this will also be true for the surrounding tissue, i.e. the gel.

Comment 3:

The temperature range (< 30°C) given at which the experiments were conducted is rather broad. Since the surrounding temperature is a crucial factor for the experiments so the exact temperature for each experimental run should be stated to allow for a correct evaluation.

Response 3:

Thank you for pointing this out. 30°C was selected as the minimum temperature to start the next experiment (naturally, the model has to cool down between each experiment). Therefore, most experiments were started at 25- 30°C. However, the first experiment of any given day for one model started at room temperature (20-25°C). We have added an additional supplementary file S3 containing the starting temperature of each experiment.

Comment 4:

Lastly, it is not clear why the measurements of each group are only performed twice. While in some groups the double measurement led to relatively similar values, there were more obvious differences in others. Although the presence of these variation in the values are discussed in the limitations & strengths section, the actual values of the single measurements are nowhere stated. They must be included in the paper. Repetition of the measurement will clarify the reason for the variation and if the reason given in the limitations & strengths section is correct. It is strongly suggested to increase the number of measurements or give a reasonable explanation for not doing so.

Response 4:

Thank you for your comment. Although the differences between the duplicate measurements could be reviewed qualitatively in Figs. 10-15 and S1, the CEM43 values of individual measurements were not shared. We have added an additional Supplementary file S3, containing an expanded version of Table 1 with individual measurements included.

The most major discrepancy between two replicates can be seen for 90°C in the model with a PMMA layer of 1mm and no PAA gel (CEM43 2E09 vs 5E07). However, both of these values are clearly higher than the upper CEM43 safe limit of 1.6E1. This was true for every measurement: In all cases, both replicates were either above or below the safe limit. Therefore, any variation in the measured CEM43 did not affect the outcomes of the study. We have added this observation to the Discussion section.

Comment 5:

Section 2.1.3 Final sample preparation: Please include some short but clear information about the used spray-paint. The given reference (15) is a paper which also gives no more detailed information but instead also references another literature which is not freely available.

Response 5:

Thank you for your comment. The spray paint we used was matte black spray paint from the brand SPECTRUM. Without the spray-paint, the thermal emissivity of the different biomaterials (Ti6Al4V, HA, PMMA) will show apparent differences in temperature due to differences in the thermal emissivity of the materials. Adding the paint equalises the emissivity. In Fig. 2 of the reference of our manuscript (https://doi.org/10.1302/2046-3758.711.BJR-2018-0080.R1) shows a side-by-side comparison of two implants with and without spray-paint, clearly exhibiting its necessity.

We have expanded Section 2.1.3. of the manuscript to clarify our use of the spray-paint.

Comment 6:

Section 2.2.2 Quantitative measurement: The exact total number of the experiments for each group has to be included.

Response 6:

We apologise for any unclarity. The number of duplicate experiments was already stated at the end of section 2.2.2.: “All experiments were conducted in duplicate and average values were reported. All experiments were carried out with and without gel phantoms.”. Since a total of 10 model systems were subjected to heating to 3 distinct temperatures in duplicate, the total number of experiments was 60. We have added this number to Section 2.2.2. to avoid confusion.

Comment 7:

Fig. 10 – 15: In the Figure captions 90°C are stated while in the figures themselves 70°C is stated. Please revise correctly. Additionally, the thermographic pictures have to be explained better, maybe by adding one schematic, especially for the upper part: what means the different height of the coloured graph? For readers not familiar with this technique this is not self-explainable.

Response 7:

Thank you for pointing out our mistake, we have changed the Figure captions accordingly.

Additionally, we agree that more clarification is needed for the cross-section profiles of the thermographs. We have added a short description to the Results section and changed

---

## [Decision Letter · Decision Letter 1]

11 Nov 2025

Dear Dr. Kamphof,

Thank you for submitting your manuscript to PLOS ONE. After careful consideration, we feel that it has merit but does not fully meet PLOS ONE’s publication criteria as it currently stands. Therefore, we invite you to submit a revised version of the manuscript that addresses the points raised during the review process.

We look forward to receiving your revised manuscript.

Kind regards,

Abbas Farmany

Academic Editor

PLOS ONE

Journal Requirements:

Reviewers' comments:

Reviewer's Responses to Questions

**Comments to the Author**

Reviewer #1: (No Response)

Reviewer #2: (No Response)

2. Is the manuscript technically sound, and do the data support the conclusions?

Reviewer #1: Yes

Reviewer #2: Yes

3. Has the statistical analysis been performed appropriately and rigorously?

Reviewer #1: Yes

Reviewer #2: N/A

4. Have the authors made all data underlying the findings in their manuscript fully available?

Reviewer #1: Yes

Reviewer #2: (No Response)

5. Is the manuscript presented in an intelligible fashion and written in standard English?

Reviewer #1: Yes

Reviewer #2: Yes

Reviewer #1: The authors corrected the paper according to the guidelines presented in the previous report, thank you for your contribution to improving the manuscript. In its present form, the paper can be further processed and accepted for publication.

Reviewer #2: Most of the comments have been answered well enough. However, not all mentioned changes in the manuscript did meet the expectations. Details and remaining advices are given below:

1. Addition of data in the supplementary material is satisfactory and appreciated.

2. Figure captions 10 – 15: Changing “thermal profile” to “thermal cross-section” may improve the understanding, however adding some information IN the figure itself is strongly recommended; at least number/letters should be included for the different figure parts (e.g. A, B, C). As I understand, the cross-section is given for the black line in the thermograph? If so, please add this information to the figure caption.

Furthermore, there is still no explanation about the “height-profile” of the cross-section. Is it just “the higher the temperature, the higher the image of cross-section”? What additional, valuable information does the cross-section offer to the reader which cannot be found in the thermograph? Please explain the reason to include the cross-section.

3. I understand the explanation for using Ttop consistently for all sample as representative to the implant – tissue – interface. However, as it is crucial for the reader to understand the terminology unmistakably and the method section for this terminology does not explain it well enough the reader relies mainly on Figure 9. Therefore, Figure 9 has to be supplemented by a second scheme for the control samples, so that it is total clear which terminology applies for which part of which sample.

4. There is a typo in the discussion (section 4.1.1, second paragraph):

“...for 3.5 minutes[14,15].[18] While…”

Please check for the use of space bar and style of given references.

**Do you want your identity to be public for this peer review?** For information about this choice, including consent withdrawal, please see our Privacy Policy

Reviewer #1: No

Reviewer #2: No

---

## [Author Response · Author response to Decision Letter 2]

19 Nov 2025

Reviewer 1

Comment 1:

The authors corrected the paper according to the guidelines presented in the previous report, thank you for your contribution to improving the manuscript. In its present form, the paper can be further processed and accepted for publication.

Response 1:

Dear reviewer, thank you for reviewing our manuscript. We are happy to hear that you are satisfied with the changes we made.

Reviewer 2

Comment 2:

Figure captions 10 – 15: Changing “thermal profile” to “thermal cross-section” may improve the understanding, however adding some information IN the figure itself is strongly recommended; at least number/letters should be included for the different figure parts (e.g. A, B, C). As I understand, the cross-section is given for the black line in the thermograph? If so, please add this information to the figure caption.

Furthermore, there is still no explanation about the “height-profile” of the cross-section. Is it just “the higher the temperature, the higher the image of cross-section”? What additional, valuable information does the cross-section offer to the reader which cannot be found in the thermograph? Please explain the reason to include the cross-section.

Response 2:

Dear reviewer, thank you for your continued scrutiny of our manuscript. Your efforts are appreciated. Indeed, the black line in the top-down thermograph indicates the position of the cross-section view.

We have changed figures 10-15 in the following ways, based on your feedback:

1. We have added letters to indicate each figure part as you suggested, and changed the Figure captions accordingly.

2. We have changed the Figure captions to explain the relation between the top-down thermograph and the thermal cross-section

The y-axis of the thermal cross-sections indicates the temperature indeed. The thermal cross-sections do not convey additional information, compared to the top-down thermographs; however, they do allow the reader to see more clearly the transitions from one part of the implant models to another, since it is generally easier to distinguish changes in shape rather than changes in hue, especially for people who are colour-blind. Since the cross-sections do not take up any space that could otherwise be utilised for something else, we have elected to keep them as-is. Please let us know if you disagree.

Comment 3:

I understand the explanation for using Ttop consistently for all sample as representative to the implant – tissue – interface. However, as it is crucial for the reader to understand the terminology unmistakably and the method section for this terminology does not explain it well enough the reader relies mainly on Figure 9. Therefore, Figure 9 has to be supplemented by a second scheme for the control samples, so that it is total clear which terminology applies for which part of which sample.

Response 3:

We agree with your suggestion to expand Figure 9 to provide more clarity on the nomenclature of the thermocouples. We have changed Figure 9 to represent all 3 configurations (bone cement, HA coating and control), and changed the figure caption accordingly.

Comment 4:

There is a typo in the discussion (section 4.1.1, second paragraph):

“...for 3.5 minutes[14,15].[18] While…”

Please check for the use of space bar and style of given references.

Response 4:

Thank you for your diligence. We have amended the error that you pointed out, as well as other formatting errors.

---

## [Decision Letter · Decision Letter 2]

20 Nov 2025

Thermal insulation of poly(methyl methacrylate) bone cement and hydroxyapatite coatings under induction heating of metal implants

PONE-D-25-36081R2

Dear Dr. Kamphof,

We’re pleased to inform you that your manuscript has been judged scientifically suitable for publication and will be formally accepted for publication once it meets all outstanding technical requirements.

Kind regards,

Abbas Farmany

Academic Editor

PLOS ONE

Additional Editor Comments (optional):

Reviewers' comments:

Reviewer's Responses to Questions

**Comments to the Author**

Reviewer #2: All comments have been addressed

2. Is the manuscript technically sound, and do the data support the conclusions?

Reviewer #2: Yes

3. Has the statistical analysis been performed appropriately and rigorously?

Reviewer #2: I Don't Know

4. Have the authors made all data underlying the findings in their manuscript fully available?

Reviewer #2: Yes

5. Is the manuscript presented in an intelligible fashion and written in standard English?

Reviewer #2: Yes

Reviewer #2: The explanations and changes to the figures are satisfactory, the readers will especially benefit from the changes made to Figure 9.

**Do you want your identity to be public for this peer review?** For information about this choice, including consent withdrawal, please see our Privacy Policy

Reviewer #2: No

---

## [Editor Report · Acceptance letter]

PONE-D-25-36081R2

PLOS ONE

Dear Dr. Kamphof,

I'm pleased to inform you that your manuscript has been deemed suitable for publication in PLOS ONE. Congratulations! Your manuscript is now being handed over to our production team.

Kind regards,

on behalf of

Dr. Abbas Farmany

Academic Editor

PLOS ONE